# Reliability of dried blood spot (DBS) cards in antibody measurement: A systematic review

**Fahimah Amini**[1]*, **Erick Auma**[2], **Yingfen Hsia**[1,3], **Sam Bilton**[4], **Tom Hall**[1], **Laxmee Ramkhelawon**[1], **Paul T. Heath**[1,4], **Kirsty Le Doare**[1,4,5]

**1** Paediatric Infectious Disease Research Group, Institute for Infection and Immunity, St. George's University of London, London, United Kingdom, **2** Department of Biology, University of Lyon, Université Claude Bernard Lyon, ENS de Lyon, CNRS, UMR, Lyon, France, **3** School of Pharmacy, Queen's University Belfast, Belfast, United Kingdom, **4** St Georges University Hospitals NHS Foundation Trust, Tooting, London, United Kingdom, **5** Pathogen Immunology Group, Public Health England, Porton Down, United Kingdom

* famini@ic.ac.uk

**Data Availability Statement:** All relevant data are within the manuscript and its Supporting information files.

## Abstract

### Background

Increasingly, vaccine efficacy studies are being recommended in low-and-middle-income countries (LMIC), yet often facilities are unavailable to take and store infant blood samples correctly. Dried blood spots (DBS), are useful for collecting blood from infants for diagnostic purposes, especially in low-income settings, as the amount of blood required is miniscule and no refrigeration is required. Little is known about their utility for antibody studies in children. This systematic review aims to investigate the correlation of antibody concentrations against infectious diseases in DBS in comparison to serum or plasma samples that might inform their use in vaccine clinical trials.

### Methods and findings

We searched MEDLINE, Embase and the Cochrane library for relevant studies between January 1990 to October 2020 with no language restriction, using PRISMA guidelines, investigating the correlation between antibody concentrations in DBS and serum or plasma samples, and the effect of storage temperature on DBS diagnostic performance.

We included 40 studies in this systematic review. The antibody concentration in DBS and serum/plasma samples reported a good pooled correlation, ($r^2 = 0.86$ (ranged 0.43 to 1.00)). Ten studies described a decline of antibody after 28 days at room temperature compared to optimal storage at -20°C, where antibodies were stable for up to 200 days. There were only five studies of anti-bacterial antibodies.

### Conclusions

There is a good correlation between antibody concentrations in DBS and serum/plasma samples, supporting the wider use of DBS in vaccine and sero-epidemiological studies, but there is limited data on anti-bacterial antibodies. The correct storage of DBS is critical and may be a consideration for longer term storage.

**Funding:** This study was funded by UK Research & Innovation (UKRI) and KLD is the grant holder. The grant number is: MR/S016570/1. URL of UKRI: https://www.ukri.org/ The funders had no role in study design, data collection and analysis, decision to publish, or preparation of the manuscript.

**Competing interests:** I have read the journal's policy and the authors of this manuscript have the following competing interests: KLD has received funding from IMmunising PRegnant women and INfants network (IMPRINT), funded by the GCRF Networks in Vaccines Research and Development which was co-funded by the MRC and BBSRC, the National Vaccine Program Office (NVPO), Bill & Melinda Gates Foundation, Grant OPP1119788, Thrasher Foundation 12250 and NIHR Imperial Biomedical Research Centre KLD2017. KLD has received an honorarium to give a seminar at Pfizer Inc. PTH is an occasional advisor to Pfizer and GSK vaccines. The remaining authors have no competing interests to declare. This does not alter our adherence to PLOS ONE policies on sharing data and materials.

## Introduction

Infectious diseases are a major global cause of morbidity and mortality affecting all age groups, especially young infants. Many of these diseases could be prevented through vaccination. However, vaccine clinical trials require blood draws from infants, which are often difficult because of both the volume required and the need for correct handling and storage of the sample [1]. This is especially true in low-income countries because of issues with cold chain maintenance and logistics of transportation from remote locations to a centralised research laboratory for processing. Detection and quantification of antibodies in the serum/plasma, offer a rapid and accurate assessment of vaccine responses. To use serum/plasma samples for serological tests, trained healthcare professionals are required to draw blood [2,3]. As well as a trained professional, there is a need for specific equipment such as vacutainers to collect whole blood and a centrifuge to separate serum/plasma from whole blood. Aside from specialised equipment, freezers are required to store the samples optimally prior to analysis. Due to the expensive nature of handling (i.e. storage, electricity) of blood samples, vaccine clinical trials are often problematic in a resource-limited setting [1]. Dried blood spots (DBS) would be a possible alternative, as they require a less complex procedure to collect than whole blood, especially from young infants [3,4].

The use of DBS as a diagnostic tool dates back to the 19th century, pioneered by Robert Guthrie for neonatal metabolic disorder screening [5]. In addition to screening for metabolic disorders, DBS cards have been utilised for human immunodeficiency virus (HIV) screening, laboratory quality control, drug testing and detection of pathogens in diverse sample types, including blood and dried plasma spots [6,7]. Numerous studies have also demonstrated that antibodies can be detected on DBS, such as in the prospective cohort study of congenital cytomegalovirus [8] and HIV infection [9]. DBS samples are cost-effective as easily portable equipment (i.e. lancet device and Whatmann 909 paper) can be used and does not require any specialist training.

Regardless of the broad use of DBS in a wide range of immunological bioanalyses, sensitivity and specificity remains uncertain regarding antibody quantification. There are no approved regulations or manufacturers' guideline on assay protocols for quantifying antibody concentration in DBS. There are also differences in DBS in terms of the cards themselves including the size and thickness of the spots and the material used to manufacture the cards. Further, there are no guidelines on how analysis should be conducted, including optimal elution methods.

This systematic review aims to assess the evidence for the use of DBS to accurately measure antibody concentrations from natural exposure and vaccination. Further, we review long-term DBS storage conditions in preparation for future sero-epidemiological or vaccine studies.

## Methodology

The protocol used for this review is registered with PROSPERO [CRD42019127840].

### Search strategy

PRISMA was used as a guideline to conduct this systematic review [10]. We searched the electronic databases Embase, Medline and Cochrane library for studies published between January 1$^{st}$, 1990 and October 15$^{th}$ 2020, comparing antibody levels in serum/plasma and DBS obtained from individuals below the age of 80 years. We additionally searched for articles describing stability of DBS at different storage temperatures over time. No language restriction was applied. The search strategy used a combination of MeSH and free terms for 'dried blood

spot' OR 'Guthrie card' AND 'antibody'. The database was last searched on the 15[th] October 2020.

The PRISMA checklist and the full search string are available in the S2 Method.

### Eligibility criteria

The studies that were considered eligible for inclusion were original research articles, concerning infectious diseases in humans, comparing antibody titres/concentrations in serum/plasma to DBS or describing stability of antibodies in DBS from longitudinal studies of storage at different temperatures. We included studies from all countries. Opinion pieces, reviews, comments, letters and conference abstracts were excluded. Studies that used animals to compare antibody levels in serum/plasma and DBS were also excluded. Studies that had insufficient data (absence of two or more of the following: number of participants, age, sensitivity, specificity, correlation of antibody levels in matched DBS-serum/plasma samples) were also excluded. Additionally, we included all studies that investigated the stability of antibodies in DBS samples.

### Study selection

Two independent reviewers (FA and EA) screened the titles and abstracts of the identified studies. After the initial screening, the reviewers obtained full texts of reports and they independently reviewed each article to determine whether it would be included in the final review. Disagreement on studies were resolved in discussion with a third reviewer (KLD).

### Data extraction

The reviewers (FA and EA) independently extracted the data from the included studies using PICO (patient, intervention, comparison and outcome) [11]. The following information was extracted from the selected studies: author, publication year, country, journal, infectious disease, aim of research, study design, duration of study, number of participants, mean age, sensitivity, specificity, method of sample collection, method of sample storage, laboratory tests used for confirmation, elution method and outcome. The country income of the included studies was classified by using their respective gross domestic product (GDP) using the World Bank [12]. All studies were either classified as a low-middle income country (LMIC) (which consisted of low-income, lower middle-income and upper middle-income countries) or as a high-income country (HIC).

### Data synthesis and analysis

Due to the high heterogeneity of study design, participants and outcomes, we were only able to conduct a narrative synthesis of included studies, summarising the findings with respect to each infectious disease. We calculated the pooled estimates of specificity, sensitivity and correlation coefficient using the accuracy data (true positive, true negative, false positive and false negative).

### Risk of bias

The risk of bias was assessed by FA using the Cochrane Risk of Bias for non-randomised studies (ROBINS-I) tool [13]. This included information on bias due to confounding, bias in selection of participants into the study, bias in classification of interventions, bias due to deviations from intended interventions, bias due to missing data, bias in measurement of outcomes and

bias in selection of the reported results. Due to the nature of the interventions considered in this review, the study's participants could not be blinded.

## Results

We identified 1,508 studies from the electronic databases published between 1[st] January 1990 to 15[th] October 2020. Using our search term, we sourced 789 papers from Medline, 667 from Embase and 52 from Cochrane. After the removal of duplicates, 837 studies were identified for abstract screening. A total of eighty-eight full text studies were assessed for eligibility, forty studies met the criteria for inclusion (Fig 1). An additional five papers investigating only antibody stability were also included.

### Study characteristics

The characteristics of the included studies are summarised in Table 1 and the antibody assessment is summarised in Table 2. Overall, DBS and serum samples from 16,255 individuals were included: 13,742 (84.5%) adults, 560 (3.4%) 5- to 17-year old's and 2,082 (12.8%) less than five years old. Two studies reported antibodies against hepatitis A [14,15], nine hepatitis B [16–18,26–28,46,48,51], ten hepatitis C [19–22,26–28,37,46,48], eight HIV [23–28,46,50], three *human papillomavirus* (HPV) [29,30,37], three measles [31,32,42], three rubella [33,34,42], two syphilis [40,46], two *H. pylori* [35,47] and two malaria [38,49]. Twelve papers reported on Chagas disease [35], Epstein-Barr virus [36], HPV, *H. pylori*, hepatitis C and polyomavirus [37], Strongyloidiasis [39], pertussis [41], hepatitis E [56], *Vibrio cholera* [2], measles, mumps and rubella [42], tuberculosis and cytomegalovirus [43], toxoplasmosis [44], trypanosoma [45], Covid-19 [52], respectively.

Twelve studies were conducted in Europe [14,18,21,27,29,30,35,41–43,50,52], four in North America [19,20,32,36], eleven in South America [15–17,22,24,34,38,39,48,49,51], seven in Africa [23,25,26,31,40,44,45] and five in Asia [28,33,37,46,47].

The risk of confounding was high in all included studies as the risks for measure of outcomes, missing data and deviation from intended interventions were unclear (S1 Table). Twenty-two of the studies had a moderate risk of bias for reporting [15,17–19,22,24,27–32,35,37,39,40,44,46–48,51,52], whereas, seventeen of the studies had a high risk of bias for reporting [14,16,20,21,23,25,26,33,34,36,38,41–43,45,49,50]. All studies had an overall high risk of bias (S1 Table). Blinding the laboratory personnel to the results of tests were not reported in any of included studies. Thirteen studies reported the stability of DBS, 35 out of the 39 studies reported the elution method and 33 out of the 39 studies reported the diagnostic performance.

### Elution method

Thirty-two (89%) reported the methodology used to elute the DBS samples [14,15,17–32,35–37,39–42,44–52]. The shortest incubation period of the DBS in elution buffer (50 μl 0.05% PBS/Tween-20) was 30 minutes [42] and the longest incubation period of the DBS in elution buffer (700 μl and 300 μl PBS/0.05% BSA) was 18–24 hours [17]) (Table 2). Nine studies used only phosphate buffered saline (DBS size ranging from 1.1-mm to 12-mm diameter, quantity of buffer ranging from 100 μl to 450 μl) to elute the DBS [17–19,25,26,32,35,37,44]. Eight studies used phosphate buffered saline with Tween (DBS size ranging from 3-mm to 12-mm diameter, quantity of buffer ranging from 50 μl to 800 μl) to elute the DBS [17,20,29,30,39,40,42,52]. Two studies compared the effect of different types of elution buffers [17,46]. Villar *et al*'s study found that DBS samples eluted in PBS/0.5% BSA had the lowest levels of non-specific reactivity in comparison to PBS alone, PBS/Tween 20 0.05%, PBS/Tween 20 0.05%/0.005% Sodium

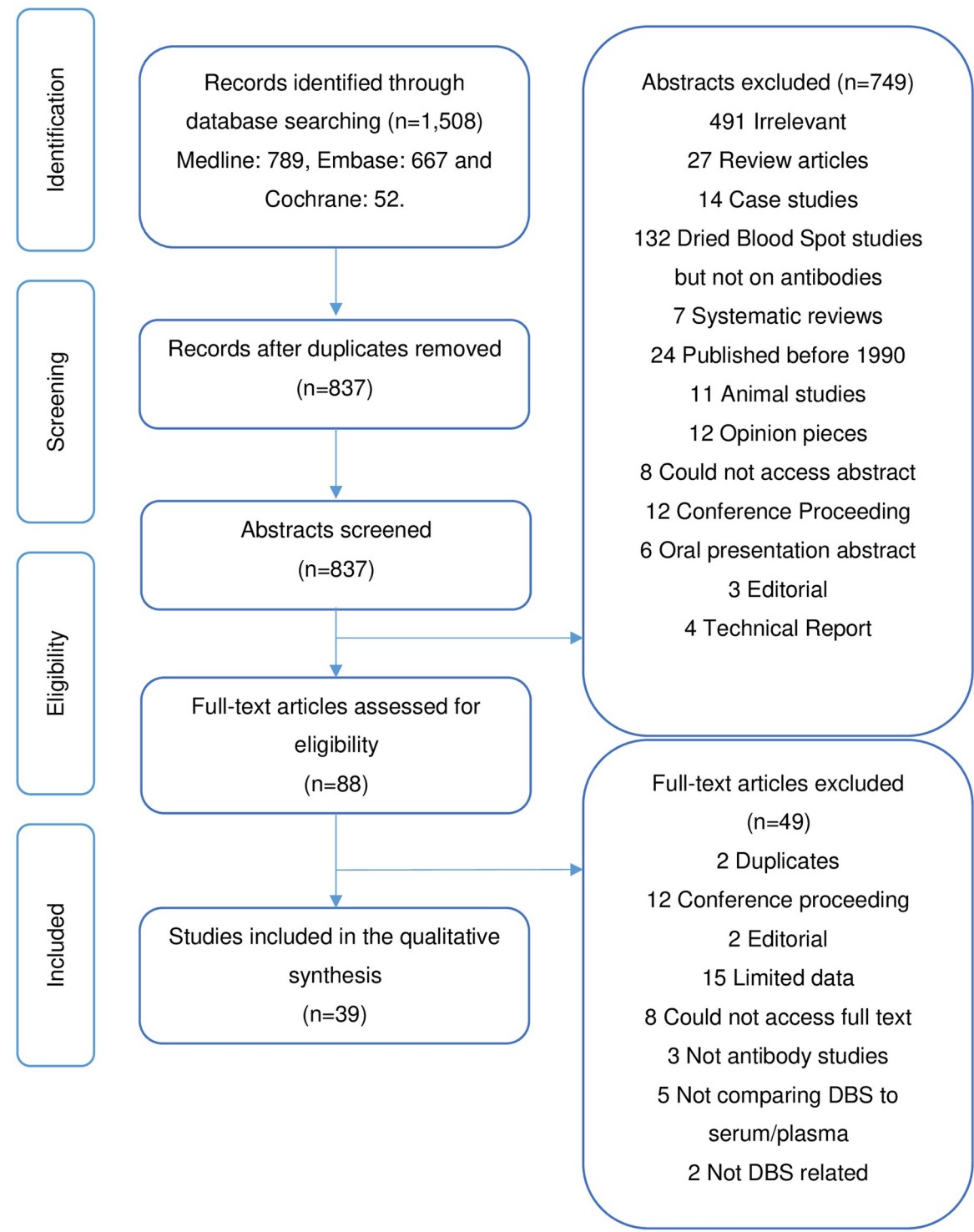

**Fig 1. Prisma flowchart.** Flowchart of studies included in the systematic review on detecting antibodies from DBS compared to venous blood samples (plasma/serum).

**Table 1. Characteristics of the included studies.**

| AUTHOR, YEAR | COUNTRY | GDP | STUDY PERIOD | STUDY DESIGN | SAMPLE SIZE | AGE | DISEASE | CONCLUSION |
|---|---|---|---|---|---|---|---|---|
| **GIL A, 1997** [14] | Spain | HIC | Nr | Cross-sectional | 298 | 15.3 ± 1.2 years | HAV | Anti-HAV antibodies are stable on DBS and ELISA is a good assay to determine anti-HAV antibodies |
| **MELGACO J, 2011** [15] | Brazil | LMIC | March 2007 | Cross-sectional | 74 | 20.6 ± 1.9 years | HAV | There is a strong correlation in anti-HAV antibodies in plasma and DBS; suggesting that commercial assays can be used to detect anti-HAV antibodies in DBS, without the need to recalculating cut-off values |
| **FLORES L, 2018** [16] | Brazil | LMIC | Nr | Cross-sectional | 155 | Adults (18 and above) | HBV | Amongst HIV positive and negative subjects, DBS could be used to quantify and detect anti-HBs |
| **VILLAR M, 2011** [17] | Brazil | LMIC | 183 days | Cross-sectional | 522 | Adults (18 and above) | HBV | Good correlation of HBV markers in serum and DBS Until 63 days, HBV markers could be detected in various temperatures, but storing DBS at -20C yielded consistent results |
| **MOHAMED S, 2013** [18] | France | HIC | Nr | Cohort | 230 | Adults (18 and above) | HBV | DBS is an alternative reliable sample to plasma specimens to quantify and detect Hepatitis B antigens |
| **DOKUBO K, 2014** [19] | USA | HIC | July 2010- June 2011 | Cross-sectional | 148 | Adults (18–30 years) | HCV | DBS and plasma showed a good correlation for anti-HCV detection |
| **TEJADA-STROP A, 2015** [20] | USA | HIC | Samples collected 5 years prior to testing | Retrospective Case-control | 33 | Adults (18 and above) | HCV | The sensitivity of anti-HCV IgG detection in stored DBS was corresponding to the matched plasma sample. |
| **TUAILLON E, 2010** [21] | France | HIC | Nr | Case-control | 200 | Adults (18 and above) | HCV | DBS is reliable as serum specimens and it can be used as an alternative to detect anti-HCV |
| **BRANDAO C, 2013** [22] | Brazil | LMIC | January 2009- March 2010 | Cross-sectional | 386 | 21–60 years | HCV | DBS is a good alternative to detect anti-HCV, although it was necessary to increase DBS sample volume due to low amount of antibody levels as compared to sera samples |

*(Continued)*

**Table 1.** (Continued)

| AUTHOR, YEAR | COUNTRY | GDP | STUDY PERIOD | STUDY DESIGN | SAMPLE SIZE | AGE | DISEASE | CONCLUSION |
|---|---|---|---|---|---|---|---|---|
| **SARGE-NIJE R, 2006** [23] | Gambia | LMIC | Nr | Cross-sectional | 200 | Adults (18 and above) | HIV | HIV test results from eluted filter paper specimen are comparable to those obtained from corresponding serum samples, depending on the strategy used. |
| **CASTRO A, 2008** [24] | Brazil | LMIC | 2005–2006 | Cross-sectional | 457 | Ranged from 2 months to 78 years | HIV | DBS has a high-performance characteristic when compared to serum specimens |
| **BOILLOT F, 1997** [25] | Sierra Leone | LMIC | April 1994 | Cross-sectional | 359 | Adults (18 and above) | HIV | DBS is feasible to perform HIV sero-surveillance in a national tuberculosis control program |
| **KANIA D, 2013** [26] | Burkina Faso | LMIC | 21st June- 1st July 2011 | Cross-sectional | 218 | 29.8 ± 11.0 years | HBV, HCV and HIV | There was good analytical performance of DBS assay obtained in this study for HCV and HBV The ODs of DBS from HIV, HCV or HBV positive subjects did not overlap with ODs of DBS samples from negative individuals |
| **ROSS S, 2013** [27] | Germany | HIC | Nr | Cross-sectional | 726 | Adults (18 and above) | HBV, HCV and HIV | There is a good correlation of antibody levels across the three diseases in matched serum and DBS. |
| **LEE C, 2011** [28] | Malaysia | LMIC | November 2009- March 2010 | Cross-sectional | 600 | Adults (18 and above) | HBV, HCV and HIV | DBS is as reliable as serum, although different cut-off points need to be used to validate test positivity in DBS |
| **BHATIA R, 2019** [29] | UK | HIC | Nr | Cross-sectional | 96 | Females (21–59 years) | HPV | A strong correlation between anti-HPV16L1 IgG and IgA was observed in serum and DBS. |
| **LOUIE K, 2018** [30] | UK | HIC | Nr | Cohort | Vaccinated: 50 Unvaccinated:103 | Vaccinated females: 21 years (median) Unvaccinated females: 26 years (median) | HPV | Although sensitivity levels of DBS are slightly lower than serum; DBS is an appropriate alternative to serum for HPV vaccine surveillance |

(*Continued*)

**Table 1.** (Continued)

| AUTHOR, YEAR | COUNTRY | GDP | STUDY PERIOD | STUDY DESIGN | SAMPLE SIZE | AGE | DISEASE | CONCLUSION |
|---|---|---|---|---|---|---|---|---|
| **UZICANIN A, 2011** [31] | Uganda | LMIC | Nr | Cross-sectional | Measles: 274 Non-measles: 75 | Measles: 31 ± 25.8 (mean months ±SD) Non-measles: 41 ± 29.5 (mean months ± SD) | Measles | Although the non-measles group had a low concordance group; the overall findings indicate that DBS are a feasible alternative to serum for serological confirmation |
| **COLSON K, 2015** [32] | Mexico and Republic of Nicaragua | LMIC | 2015 | Cross-sectional | Mexico:1134 Nicaragua:454 | Mexico-17.5 ± 0.19 (months mean ± SE) Nicaragua-17.7 ± 0.23 (months mean ± SE) | Measles | It is feasible to conduct sero-surveys using DBS samples in low-resource settings, as the antibody levels are highly consistent in matched serum and DBS samples |
| **PUNNARUGSA V, 1991** [33] | Thailand | LMIC | Nr | Cross-sectional | 1000 | Adults (18 and above) | Rubella | 80.7% of DBS had similar antibody titres to matched serum. |
| **HELFAND F, 2007** [34] | Peru | LMIC | June 2004- May 2005 | Cross-sectional | 376 | Children (8 months and above) | Rubella | There was a good correlation between serum and DBS; which suggests that DBS is an acceptable to be used as a sample for serological tests |
| **HOLGUIN A, 2013** [35] | Spain | HIC | May 2011- March 2012 | Cross-sectional | 147 | Adults (18 and above) | Chagas | With a cut-off point of ≥0.88, there are low numbers of false-negative DBS |
| **EICK G, 2017** [36] | USA | HIC | November 2014-February 2015 | Cohort | 208 | Adults (18–77 years) | Epstein-Barr virus | This study showed that there is a high correlation of Epstein-Barr virus antibodies in plasma and DBS. |
| **WATERBOER T, 2012** [37] | Mongolia | LMIC | Nr | Cross-sectional | 985 | Females (16–63 years) | HPV, *H. pylori*, HCV and Polyomavirus | There was a good correlation between serum and DBS in high-titre antibodies when compared to low-titre antibodies |
| **DUARTE E, 2002** [38] | Brazil | LMIC | September 1996 | Retrospective | 210 | 1–64 years (25.3 years) | Malaria | DBS is as reliable as serum samples to determine malaria status in patients |
| **FORMENTI F, 2016** [39] | Ecuador | LMIC | Nr | Cross-sectional | 235 (174 children and 61 adults) | Adults (19–70 years) Children (7–16 years) | Strongyloidiasis | Good concordance between standard serology and DBS results |
| **SMIT P, 2013** [40] | Tanzania | LMIC | Nr | Cohort | 1645 | 15–84 years (31.9 years) | Syphilis | It is recommended that Treponema pallidum particle agglutination assay is used to quantify antibody levels in DBS |

(*Continued*)

**Table 1.** (Continued)

| AUTHOR, YEAR | COUNTRY | GDP | STUDY PERIOD | STUDY DESIGN | SAMPLE SIZE | AGE | DISEASE | CONCLUSION |
|---|---|---|---|---|---|---|---|---|
| **VAN OMMEN C, 2012** [41] | Netherlands | HIC | August 2006-April 2007 | Cross-sectional | 129 | Mothers and new-born's | Pertussis | Anti-PT IgG from cord blood in DBS and serum showed a high coefficient of correlation |
| **CONDORELLI F, 1994** [42] | Italy | HIC | January 1993-March 1993 | Cross-sectional | 228 | Children and adults | Measles, mumps and rubella | (1) Antibodies are stable for at least for 2 weeks at room temperature (2) The antibody levels DBS and serum showed a good agreement as they were higher than 96% |
| **AABYE M, 2012** [43] | Finland | HIC | Nr | Cross-sectional | 52 | Adults (18 and above) | Tuberculosis and Cytomegalovirus | Correlation of IP-10 marker was excellent in DBS and plasma samples |
| **HEGAZY M, 2020** [44] | Egypt | LMIC | Nr | Cross-sectional | 101 | 10–70 years | Toxoplasmosis | DBS can be used to test for *Toxoplasma* as it has high diagnostic accuracy |
| **GEERTS M, 2020** [45] | Democratic Republic of Congo | LMIC | Nr | Cross-sectional | 132 | Nr | Trypanosoma brucei | The high sensitivity and specificity suggests that DBS can be used determine *Trypanosoma brucei* infection status |
| **MA J, 2020** [46] | China | HIC | Nr | Cross-sectional | 429 | Adults (18 and above) | HIV, Hepatitis C and syphilis | The antibody concentration in DBS samples were comparable to the antibody concentration in the matched plasma samples |
| **KUMAR A, 2019** [47] | India | LMIC | January 2018-May 2018 | Observational | 88 | Adults (18 and above) | *H. pylori* | No difference in antibody levels were seen in matched DBS and plasma samples |
| **VILLAR L, 2020** [48] | Argentina | HIC | October 2014-December 2014 | Cross-sectional | 622 | Adults (36.6 ±14.3) | HBV and HCV | Although the sensitivity % for anti-HBc, DBS samples still can be utilised for detecting anti-HBV and anti-HCV antibody concentrations |
| **ROSAS-AGUIRRE A, 2020** [49] | Peru | HIC | August 2012-September 2012 | Longitudinal cohort | 470 | Adults (18 and above) | *Plasmodium vivax* | To detect anti-P.vivax antibodies, it is better to use serum rather than DBS samples |
| **STEFIC K, 2019** [50] | France | HIC | Nr | Cross-sectional | 10 | Adults (18 and above) | HIV | This study is further showing evidence that DBS samples are also suitable for detecting HIV antibodies, as well as, for use in population surveillance |

(*Continued*)

**Table 1.** (Continued)

| AUTHOR, YEAR | COUNTRY | GDP | STUDY PERIOD | STUDY DESIGN | SAMPLE SIZE | AGE | DISEASE | CONCLUSION |
|---|---|---|---|---|---|---|---|---|
| **CRUZ H, 2020** [51] | Brazil | LMIC | 2009–2014 | Cross-sectional | 2309 | Adults (18 and above) | HBV | HBsAg in DBS samples had the best sensitivity percentage in this study. This study showed that DBS samples can be utilised for detecting HBV infections |
| **MORLEY G, 2020** [52] | UK | HIC | 2020 | Cross-sectional | 87 | Adults (18 and above) | Coronavirus | The results of this study illustrate that antibodies in DBS samples are highly comparable to serum samples |

Abbreviations: [a] GDP: Gross domestic product; [b] HIC: High Income Country; [c] LMIC: Lower Middle-Income Countries; [d] HAV: Hepatitis A virus; [e] HBV: Hepatitis B virus; [f] HCV: Hepatitis C virus; [g] HIV: Human Immunodeficiency virus; [h] HPV: Human papillomavirus; [i] anti-HBs: Hepatitis B surface antibody; [j] Nr: Not reported.

azide and PBS/Tween 20 0.2%/5% BSA. Whereas, Ma *et al* found that eluting DBS spots in 500 μl of 1%Tween-20/PBS resulted in the highest antibody recovery.

## Diagnostic performance

Thirty of the studies used enzyme-linked immunosorbent assays (ELISA) [14–17,19,21–26,29–32,34,36,38,39,41–47,49–52] with the one study using the Luminex 100 [37] or a combination of Treponema pallidum particle agglutination assay (TPPA), Treponema pallidum hemagglutination assay (TPHA) and ELISA [40]. Two studies used the architect system [27,35] as a detection method and two studies used chemiluminescence immunoassay (CIA) either alone [18] or in combination with ELISA [20]. The method of antibody quantification was unclear in one study [35] and not reported in another study [28] (Table 2).

Thirty-three of the included studies reported the sensitivity [14–28,30–35,37,40,42,44–52] and twenty-six reported the specificity [14,15,17–28,30–33,35,40,44–49,51,52] of antibodies on DBS (Fig 2). The pooled sensitivity for all the infectious diseases ranged from 35.2% to 100% with a mean of 98.8% and the pooled specificity ranged from 50.4% to 100% with a mean of 95.4%. The highest mean sensitivity and specificity reported were for HIV; 97.5% and 99.6%, respectively [23–28].

The lowest mean sensitivity reported were for the malaria (*P. vivax*) study; 50% [49] whereas the lowest specificity reported were for syphilis; 50.4% (ELISA) [40] (Table 3).

Nineteen studies reported the correlation between antibody concentration in DBS and serum or plasma samples [13,15–17,20,24,26,28–30,32–37,39]. The pooled correlation of antibody concentration in DBS and serum/plasma samples ranged from 0.43 to 1.00 with a mean of 0.86. The highest correlation of antibody titers in DBS and serum/plasma samples was observed for Coronavirus, which was 0.97 [52]. The lowest mean correlation of DBS and serum/plasma were observed for the measles study; 0.33 [31] (Table 3).

## Storage of DBS cards

Twenty-one studies reported that their DBS samples were stored at -20˚C [17,19,22–24,26,27,30,33,34,36,39,41–43,46,48–50,53], three studies stored at -80˚C [37,38,44], five studies stored at 4˚C [16,31,32,44,51], two studies stored at room temperature [20,35] and in two studies the DBS's were stored at -5˚C to -10˚C [40] and -70˚C [21], respectively (Table 2).

**Table 2. Immunological assessment of the included studies.**

| AUTHOR, YEAR | TYPE OF SCREENING | LABORATORY TEST USED TO CONFIRM PATHOGEN | ELUTION METHOD OF DBS | SAMPLE STORAGE |
|---|---|---|---|---|
| **GIL A, 1997** [14] | DBS and serum | ELISA | DBS were eluted for 12 hours at room temperature in 1 ml of 0.115% saline/1.5% bovine albumin | DBS: 4˚C; Serum: -20˚C |
| **MELGACO J, 2011** [15] | DBS and plasma | ELISA | 12.5mm-diameter DBS were eluted at 4˚C overnight in 350 µl PBS/0.2% Tween-20/5% bovine plasma albumin | -20˚C |
| **FLORES L, 2018** [16] | DBS and serum | ELISA | Nr | Nr |
| **VILLAR M, 2011** [17] | DBS and serum | ELISA | 6-mm DBS samples were eluted in 700 µl of PBS/0.5% BSA for HBsAg and anti-HBc antibody quantification. 12.5mm DBS samples were eluted in 300 µl of PBS/0.5% BSA for HBsAg and anti-HBc antibody quantification. All DBS samples were incubated at 2˚-6˚C for 18–24 hours. | -20˚C |
| **MOHAMED S, 2013** [18] | DBS and plasma | CIA | 12-mm diameter DBS were eluted at room temperature in continued agitation in 450 µl PBS | DBS: RMT; Serum: Nr |
| **DOKUBO K, 2014** [19] | DBS and serum | ELISA | 6.35-mm diameter DBS were eluted overnight at 2˚-8˚C in 125 µl PBS | -70˚C |
| **TEJADA-STROP A, 2015** [20] | DBS and plasma | CIA and ELISA | 12-mm diameter DBS were eluted overnight at 4˚C in 500 µl PBS/Tween | -20˚C for 5 years |
| **TUAILLON E, 2010** [21] | DBS and serum | ELISA | 6-mm diameter DBS were eluted on shaker overnight in 200 µl ELISA kit sample diluent | DBS: -20˚C; Serum: Nr |
| **BRANDAO C, 2013** [22] | DBS and serum | ELISA | 3-mm diameter DBS were eluted for 18–24 hours at 4˚-8˚C in 300 µl PBS/0.5% BSA | -20˚C |
| **SARGE-NIJE R, 2006** [23] | DBS and serum | ELISA | 5.5-mm diameter DBS eluted overnight at 4˚C in 100 µl or 200 µl PBS/Tween-20/0.005% Sodium azide | Nr |
| **CASTRO A, 2008** [24] | DBS and serum | ELISA | 4.7-mm diameter DBS eluted in ELISA kit sample diluent | DBS: -20˚C; Serum: Nr |
| **BOILLOT F, 1997** [25] | DBS and serum | ELISA | DBS in PBS was placed on rotatory shaker for 90 mins, followed by overnight incubation at 4˚C | -20˚C |
| **KANIA D, 2013** [26] | DBS and plasma | ELISA | 6-mm diameter DBS were eluted overnight in 300 µl (HBsAg and HCV) or 150 µl (anti-HBc) PBS | Nr |
| **ROSS S, 2013** [27] | DBS and serum | Architect system | DBS were eluted overnight at room temperature on a shaker in 1 ml PBS/0.05% Tween-20/0.08% Sodium azide | Nr |
| **LEE C, 2011** [28] | DBS and plasma | Nr | 5.5-mm diameter DBS were eluted for 1 hour at 4˚C in 500 µl miliQ water | DBS: -20˚C; Serum: -80˚C |
| **BHATIA R, 2019** [29] | DBS and serum | ELISA | DBS were eluted in 800 µl PBS/0.05% Tween-20 | DBS: 4˚C; Serum: Nr |
| **LOUIE K, 2018** [30] | DBS and serum | ELISA | 9x 2-mm diameter DBS were eluted overnight at 4˚C on a shaker in 200 µl PBS/0.05% Tween-20 | 4˚C |
| **UZICANIN A, 2011** [31] | DBS and serum | ELISA | 6.35-mm diameter DBS were eluted in 250 µl PBS/Tween-20/5% non-fat dry milk followed by placing it on a plate shaker for 30 mins. Following on, the samples were incubated for 16 hours at 4˚C | -20˚C |
| **COLSON K, 2015** [32] | DBS and serum | ELISA | 6-mm diameter DBS were eluted for 14–16 hours at 6˚-8˚C in 400 µl PBS | -20˚C |
| **PUNNARUGSA V, 1991** [33] | DBS and serum | Unclear | Unclear | DBS: RMT; Serum: -20˚C |
| **HELFAND F, 2007** [34] | DBS and serum | ELISA | Nr | -20˚C |
| **HOLGUIN A, 2013** [35] | DBS and serum | Architect Chagas assay | 2x 1.1-mm diameter DBS were eluted overnight at room temperature on a shaker in 300 µl PBS | -80˚C |
| **EICK G, 2017** [36] | DBS and plasma | ELISA | 3.2-mm diameter DBS were eluted overnight at 4˚C on a shaker in 250 µl ELISA kit sample diluent | -80˚C |
| **WATERBOER T, 2012** [37] | DBS and serum | Multiplex serology (Luminex 100 analyser) | 2.85-mm diameter DBS were eluted overnight at 4˚C in 100 µl PBS | -20˚C |

(*Continued*)

**Table 2.** (Continued)

| AUTHOR, YEAR | TYPE OF SCREENING | LABORATORY TEST USED TO CONFIRM PATHOGEN | ELUTION METHOD OF DBS | SAMPLE STORAGE |
|---|---|---|---|---|
| **DUARTE E, 2002** [38] | DBS and serum | ELISA | Unclear | DBS: -5°C to -10°C Serum: -70°C |
| **FORMENTI F, 2016** [39] | DBS and serum | ELISA | 8 DBS were eluted overnight at room temperature in PBS/Tween-20 | -20°C |
| **SMIT P, 2013** [40] | DBS and plasma | TPPA, ELISA and TPHA | 6-mm diameter DBS were eluted overnight at 4°C in 100 μl PBS/ 0.05% Tween-80 | -20°C |
| **VAN OMMEN C, 2012** [41] | DBS and serum | ELISA | 2x 3.18-mm diameter DBS were vortexed at room temperature for 1 hour in 400 μl 0.5%BSA/0.01% Tween-20 | -20°C |
| **CONDORELLI F, 1994** [42] | DBS and serum | ELISA | DBS were eluted at room temperature for 30 mins in agitation in 50 μl 0.05% PBS/Tween-20 | DBS: 4°C; Serum: -20°C |
| **AABYE M, 2012** [43] | DBS and plasma | ELISA | Unclear | -80°C |
| **HEGAZY M, 2020** [44] | DBS and serum | ELISA | DBS were eluted in 0.2mL PBS at room temperature overnight on an automatic shaker. Samples were then centrifuged at 1000xg for 10 minutes | -20°C |
| **GEERTS M, 2020** [45] | DBS and plasma | ELISA | 6-mm DBS spots were eluted overnight in 400 μl *g*-iELISA sample diluent | Nr |
| **MA J, 2020** [46] | DBS and plasma | ELISA | DBS samples were eluted for comparison for all three antibodies in 1%Tween-20/PBS, 1%TritonX100/PBS, 1%Tween-20 and 1% TritonX100. Furthermore, the elution effect of Tween-20/PBS buffer was compared at 1% and at 3% | DBS: -20°C; Plasma: -20°C |
| **KUMAR A, 2019** [47] | DBS and plasma | ELISA | 6-mm DBS spot was eluted in 500 μl of diluent buffer (part of the ELISA kit) for 2–3 hours at room temperature on a plate shaker | DBS: -20°C; Plasma: -80°C |
| **VILLAR L, 2020** [48] | DBS and serum | PCR | For anti-HCV, a 3-mm spot was placed into a microtube containing 300 μl of PBS/0.5%BSA at 4–8°C for 18–24 hours. A 6-mm spot was eluted in 700 μl of PBS/0.5%BSA at 4–8°C for 18–24 hours for HBsAg and anti-HBc detection. | -20°C |
| **ROSAS-AGUIRRE A, 2020** [49] | DBS and serum | ELISA | 5-mm disc was eluted overnight at 4°C in PBS/non-fat milk/0.05% Tween-20 | DBS: 4°C; Serum: -70°C |
| **STEFIC K, 2019** [50] | DBS and plasma | ELISA | 6-mm spot was placed in 150 μl 0.01M-PBS/10%BSA/0.05% Tween-20 and it was incubated overnight at 4°C | Nr |
| **CRUZ H, 2020** [51] | DBS and serum | ELISA | To detect HBsAg and total anti-HBc, a 6-mm DBS spot was eluted in 700 μl PBS/0.05%BSA. For anti-HBc detection, a 12.5 mm spot was eluted in 300 μl of PBS/0.05%BSA. | -20°C |
| **MORLEY G, 2020** [52] | DBS and serum | ELISA | One DBS spot was eluted in 250 μl of 0.05% PBS/Tween-20. The tubes were briefly vortexed before it was incubated overnight at toon temperature. On the following day the tubes were centrifuged for 10 minutes at 10,600 xg at room temperature. | Nr |

Abbreviations: [a] HAV: Hepatitis A virus; [b] HCV: Hepatitis C virus; [c] HIV: Human Immunodeficiency virus; [d] HPV: Human papillomavirus; [e] HBsAg: Hepatitis B surface antigen; [f] anti-HBc: Hepatitis B core antibody; [g] anti-HBs: Hepatitis B surface antibody; [h] BSA: Bovine serum albumin; [i] CIA: Chemiluminescence immunoassay; [i] PBS: Phosphate buffered saline; [j] PCR: Polymerase chain reaction; [k]TPPA: Treponema pallidum particle agglutination assay; [l] TPHA: Treponema pallidum hemagglutination assay; [m] Nr: Not reported; [n] RMT: Room temperature.

Thirteen studies investigated the stability of antibodies in DBS samples stored at different temperatures (Table 4). Six studies [2,21,23,35,55] concluded that antibody levels in DBS were stable at room temperature, ranging from 7 days to 28 days. A slight decline was observed in antibody concentrations in DBS samples that were stored at 2–8°C, although five studies [2,26,46,55,58] demonstrated that DBS samples were stable for up to 210 days at this temperature (range of storage time: 7 to 210 days). Five studies [26,38,55,57,58] found that antibodies in DBS samples stored at 37°C were unstable with antibody concentrations steadily declining in as few as three days [55]. One study showed that antibodies in DBS samples stored at 37°C

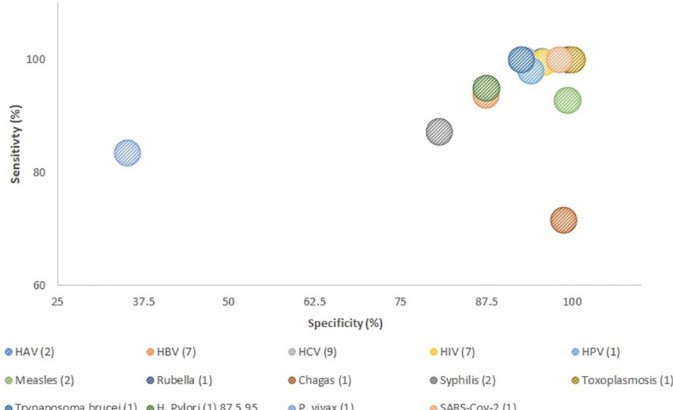

**Fig 2. Scatterplot of diagnostic performances (sensitivity and specificity) based on pathogen type.**

were stable until the 7th day [2]. Nine studies [2,19,24,26,38,46,55–57] demonstrated minimal variations in antibody concentrations compared to baseline cards stored at -20˚C, over 21 to 200 days.

## Storage of DBS cards

Twenty-one studies reported that their DBS samples were stored at -20˚C [15,17,20–22,24,25,28,31,32,34,37,39–41,44,46–48,51], three studies stored at -80˚C [35,36,42], five studies stored at 4˚C [14,29,30,42,49], two studies stored at room temperature [18 = 33] and in two studies the DBS's were stored at -5˚C to -10˚C [38] and -70˚C [19], respectively (Table 4).

Thirteen studies investigated the stability of antibodies in DBS samples stored at different temperatures (Table 4). Six studies [2,19,21,33,54] concluded that antibody levels in DBS were stable at room temperature, ranging from 7 days to 28 days. A slight decline was observed in antibody concentrations in DBS samples that were stored at 2–8˚C, although five studies [2,24,44,53,56] demonstrated that DBS samples were stable for up to 210 days at this temperature (range of storage time: 7 to 210 days). Five studies [24,36,53,55,56] found that antibodies in DBS samples stored at 37˚C were unstable with antibody concentrations steadily declining in as few as three days [53]. One study showed that antibodies in DBS samples stored at 37˚C were stable until the 7th day [2]. Nine studies [2,17,22,24,36,44,53–55] demonstrated minimal variations in antibody concentrations compared to baseline cards stored at -20˚C, over 21 to 200 days.

## Discussion

To our knowledge, this is the first comprehensive systematic review to summarise the utility of DBS considering the key aspects of storage, assay methods and card handling which are all important considerations for vaccine trials and serological studies. Overall, the diagnostic accuracy and precision was high when comparing serum/plasma to DBS, indicating that DBS are a useful alternative to serum.

In a review of anti-HCV antibodies eluted from DBS, Vazques-Moron *et al* reported a sensitivity of >96% and a specificity of >99% for anti-HCV antibodies in DBS samples [57]. Their reported figures are similar to the pooled diagnostic performances we have shown. However, our pooled results indicate that there may be differences in both sensitivity and specificity depending on the pathogen type. A study of SARS-COV-2 antibodies demonstrated that sensitivity of matched plasma and DBS was 98.9% [52]. This is potentially useful knowledge during

**Table 3. Sensitivity, specificity and $r^2$ of the included studies.**

| AUTHOR, YEAR | SENSITIVITY | SPECIFICITY | CORRELATION ($R^2$) OF ANTIBODY LEVELS IN DBS AND SERUM/PLASMA |
|---|---|---|---|
| **GIL A, 1997** [14] | 91.30% | 99.30% | Nr |
| **MELGACO J, 2011** [15] | 100% | 100% | 0.6262 |
| **FLORES L, 2018** [16] | Anti-HBs in HIV- individuals: 79.8% Anti-HBs in HIV+ individuals: 76.8% | Nr | Nr |
| **VILLAR M, 2011** [17] | HBsAg: 95.45%; Anti-HBc: 89.19%; Anti-HBs: 83.05% | HBsAg: 70.79%; Anti-HBc: 97.53%; Anti-HBs: 81.33% | 0.832 |
| **MOHAMED S, 2013** [18] | 98% | 100% | 0.98 |
| **DOKUBO K, 2014** [19] | Anti-HCV: 70% | Anti-HCV: 100% | 0.69 |
| **TEJADA-STROP A, 2015** [20] | CIA- 100%; ELISA- 97% | CIA- 100%; ELISA- 100% | Nr |
| **TUAILLON E, 2010** [21] | 99% | 98% | Nr |
| **BRANDAO C, 2013** [22] | 95.00% | 100% | 0.971 |
| **SARGE-NIJE R, 2006** [23] | Ranged from 95% to 100% [depending on which assay was used] | Ranged from 97.5% to 100% [depending on which assay was used] | Nr |
| **CASTRO A, 2008** [24] | 100% | 99.60% | Nr |
| **BOILLOT F, 1997** [25] | 87.50% | 100% | Nr |
| **KANIA D, 2013** [26] | HBsAg: 96%; Anti-HBc: 99.3%; HIV: 100%; HCV: 100% | HBsAg: 100%; Anti-HBc: 98.7%; HIV: 100%; HCV: 100% | HBsAg: 0.98; Anti-HBc: 0.98; HIV: 1.00; HCV: 1.00 |
| **ROSS S, 2013** [27] | HBsAg: 98.6%; Anti-HBc: 97.1%; Anti-HBs: 97.5%; Anti-HCV: 97.8%; Anti-HIV ½: 100% | HBsAg: 100%; Anti-HBc: 100%; Anti-HBs: 100%; Anti-HCV: 100%; Anti-HIV ½: 100% | Nr |
| **LEE C, 2011** [28] | HIV: 100%; HBsAg: 96.5%; Anti-HBs: 74.2%; Anti-HCV: 100% | HIV: 100%; HBsAg: 97.8%; Anti-HBs: 86.9%; Anti-HCV: 100% | HIV Ag/Ab: 0.824; HBsAg: 0.432; Anti-HBs: 0.721; Anti-HCV: 0.631 |
| **BHATIA R, 2019** [29] | Nr | Nr | Nr |
| **LOUIE K, 2018** [30] | 94% | 98% | 0.961 |
| **UZICANIN A, 2011** [31] | Measles (week 1 after rash): 98.7% Non-measles: 50% | Measles (week 1 after rash): 88.9% Non-measles: 100% | Measles (week 1 after rash): 0.71 Non-measles: 0.33 |
| **COLSON K, 2015** [32] | 100% | 96.8% | 0.92 |
| **PUNNARUGSA V, 1991** [33] | 99.40% | 100% | Nr |
| **HELFAND F, 2007** [34] | IgM: 82% IgG: 89% | Nr | IgM: 0.91 IgG: 0.94 |
| **HOLGUIN A, 2013** [35] | 98.80% | 71.60% | 0.803 |
| **EICK G, 2017** [36] | Nr | Nr | 0.93 |
| **WATERBOER T, 2012** [37] | HPV: 94.5%; *H. pylori*: 96.6%; HCV: 98%; Polyomavirus: 98% | Nr | *H. pylori*, HCV, JCV: 0.88; HPV: 0.79 |
| **DUARTE E, 2002** [38] | Nr | Nr | 0.95 |
| **FORMENTI F, 2016** [39] | Nr | Nr | 0.921 |
| **SMIT P, 2013** [40] | TPPA: 85.4% TPHA: 50.5% ELISA: 94.6% | TPPA: 98.9% TPHA: 99.7% ELISA: 50.4% | Nr |
| **VAN OMMEN C, 2012** [41] | Nr | Nr | 0.91 |
| **CONDORELLI F, 1994** [42] | Measles: 98.6%; Mumps: 96%; Rubella: 99.1% | Nr | Nr |
| **AABYE M, 2012** [43] | Nr | Nr | Nr |
| **HEGAZY M, 2020** [44] | 100% | 100% | Nr |
| **GEERTS M, 2020** [45] | 92.6% | 100% | Nr |
| **MA J, 2020** [46] | Anti-HCV: 98.32% Anti-HIV: 88.46% Anti-TP: 92.19% | Anti-HCV: 100% Anti-HIV: 100% Anti-TP: 100% | Anti-HCV: 0.99 Anti-HIV: 0.96 Anti-TP: 0.95 |

*(Continued)*

**Table 3.** (Continued)

| AUTHOR, YEAR | SENSITIVITY | SPECIFICITY | CORRELATION ($R^2$) OF ANTIBODY LEVELS IN DBS AND SERUM/PLASMA |
|---|---|---|---|
| **KUMAR A**, [47] | Anti-*H. pylori*: 87.5% | Anti-*H. pylori*: 95% | Nr |
| **VILLAR L, 2020** [48] | HBsAg: 100%; Anti-HBc: 66.6%; Anti-HCV: 75% | HBsAg: 98.9%; Anti-HBc: 99.8%; Anti-HCV: 99.8% | Nr |
| **ROSAS-AGUIRRE A, 2020** [49] | 35.2% | 83.5% | Nr |
| **STEFIC K, 2019** [50] | 98.9% | Nr | Nr |
| **CRUZ H, 2020** [51] | HBsAg: 81.2% Anti-HBc: 66.5% Anti-HBs: 60% | HBsAg: 93.5% Anti-HBc: 99.3% Anti-HBs: 75.1% | Nr |
| **MORLEY G, 2020** [52] | 98.11% | 100% | 0.975 |

Abbreviations: [a] anti-HBc: Hepatitis B core antibody; [b] anti-HBs: Hepatitis B surface antibody; [c] CIA: Chemiluminescence immunoassay; [d] ELISA: Enzyme-linked immunosorbent assay; [e] HAV: Hepatitis A virus; [f] HCV: Hepatitis C virus; [g] HIV: Human Immunodeficiency virus; [h] HPV: Human papillomavirus; [i] HBsAg: Hepatitis B surface antigen [j] Nr: Not reported; [k] TPPA: Treponema pallidum particle agglutination assay; [l] TPHA: Treponema pallidum hemagglutination assay.

the pandemic as DBS could be used as an alternative to blood samples for national surveillance. DBS sampling would be more convenient for sampling as it does not require attendance at clinic to collect samples and samples can be easily sent back by post.

There are no guidelines on how DBS samples should be stored for short- and long-periods of time and this is evident from the variable storage described in the studies we reviewed. We have demonstrated that storage at room temperature (22–28˚C) is acceptable for up to 28 days; making the transportation of DBS samples straightforward, especially in environments lacking cold chain. However, we also report that longer term storage should be at refrigerated or frozen temperatures after 28 days at room temperature, as antibodies degraded significantly thereafter. Overall, our results indicate that -20˚C is the optimum temperature to store DBS samples for prolonged periods and it may be necessary for this to be factored into trials where samples may be stored for several years prior to use. This is consistent with the national laboratory guidelines in Denmark [58], Scotland [59], US [60] and Germany [61], which all recommend long term freezer storage. Williams *et al* [62] re-quantified anti-HIV antibodies in four high positive controls that were initially spotted 23 years prior to the analysis. The antibody concentrations obtained were the same as those measured 23 years prior when stored at -20˚C. Yel *et al's* [44] study found that IgA antibodies were stable for up to 14 days at room temperature and at 4–8˚C, however, IgA antibodies were stable for up to 10 days when stored at -20˚C. Going forward, it may be useful to determine the stability of the different antibody isotypes, as certain antibodies may be more stable than others on DBS.

Whilst this review provides support for the use of DBS for the investigation of immunity to several pathogens, we found only four studies which investigated the antibody concentration against bacterial infections. It is vital that more research is undertaken to understand the stability of antibodies against bacterial infections in DBS samples and how antibody concentrations compare to serum or plasma samples. This is of particular importance if DBS are to be used in vaccine trials against bacterial pathogens, which are required to reduce the impact of the continued spread of antibiotic resistance.

There is of considerable urgency as many bacterial infections are becoming increasingly antibiotic resistant and DBS could be used as part of studies to measure vaccine or natural immunity to bacterial infections [63].

**Table 4. Optimal temperature for dried blood spot storage.**

| AUTHOR, YEAR | TEMPERATURE | DAYS | OPTIMUM TEMPERATURE TO STORE DBS FOR MAXIMUM ANTIBODY RECOVERY |
|---|---|---|---|
| **IYER A, 2018** [2] | 37˚C (humid), 37˚C (non-humid), Rmt (~22˚C), 4˚C, -20˚C | 7 | The vibriocidal antibodies on DBS samples were stable across all temperatures for one week. |
| **VILLAR M, 2011** [17] | Rmt (22˚C to 25˚C bag), Rmt (22˚C to 25˚C), 4˚C to 8˚C, -20˚C | 183 | For the first 63 days, samples (non-reactive DBS samples to anti-HBc markers) stored in all the temperatures (4˚C, -20˚C and room temperature) were stable. However, on day 183, samples stored at -20˚C had the lowest variation of optical density values. Reactive DBS samples (HBsAg marker) that were stored at -20˚C were the only ones which did not result in false-positive. The DBS samples (HBsAg marker) stored at Rmt became ELISA negative on the 63rd day. DBS samples (anti-HBs) stored at Rmt resulted in one false-positive result on the 183rd day. Nevertheless, the DBS samples (anti-HBs) stored at -20˚C had the lowest OD variation. |
| **MOHAMED S, 2013** [19] | Rmt | 14 | DBS samples (HBsAg and anti-HBs markers) stored at room temperature were stable for up to 14 days. |
| **TUAILLON E, 2010** [21] | Rmt | 12 | DBS samples (anti-HCV marker) stored at room temperature were stable for as long as 12 days. |
| **BRANDAO C, 2013** [22] | Rmt (22˚C to 26˚C), 2˚C to 8˚C, -20˚C | 117 | DBS samples (anti-HCV marker) stored at Rmt observed a significant decline in mean optical density on the 117th day in comparison to the baseline. The lowest optical density variation was observed in DBS samples stored at -20˚C until day 117th days. |
| **CASTRO A, 2008** [24] | 37˚C, Rmt (mean ~22.1˚C), 4˚C, -20˚C, -70˚C | 42 | DBS samples (anti-HIV1 marker) stored at 37˚C degraded after the 4th week. The samples stored at Rmt were stable on the 1st week, however, on the sixth week one samples which was HIV-positive (with a low optical density) became HIV-negative. The DBS samples stored at 4˚C, -20˚C and -70˚C were stable until the 6th week. |
| **PUNNARUGSA V, 1991** [33] | Rmt | 28 | Antibodies in DBS samples stored at room temperature were stable for as long as 28 days. |
| **EICK G, 2017** [36] | 37˚C, Rmt, -20˚C | 21 | DBS stored (EBV markers) at 37˚C, room temperature and -20˚C were stable for 21 days. Antibodies in DBS samples stored at 37˚C decreased over time. Recovery of antibodies in DBS samples stored at -20˚C had no clear change; indicating that -20˚C is the optimum storage temperature for DBS. |
| **HEGAZY M, 2020** [44] | 4˚C, -20˚C | 4˚C: 1 month -20˚C: 3 months | Antibodies in DBS samples stored at 4˚C were stable for one month and in samples that were stored at -20˚C for three months. |
| **YEL L, 2015** [53] | 36˚C to 38˚C, Rmt, 2˚C to 8˚C, -20˚C to -40˚C | 14 | IgG and IgM antibodies in DBS samples were stable at Rmt, 2˚C to 8˚C and -20˚C to -40˚C until the 14th day. However, IgG and IgM antibodies in DBS samples stored at 36˚C to 38˚C resulted in a decrease of antibody concentration after the 4th day. On the other hand, IgA antibodies in DBS samples stored at Rmt and 2˚C to 8˚C were stable until the 14th day. The DBS samples (IgA) stored at -20˚C to -40˚C were only stable until the 10th day, whereas the samples which were stored at 36˚C to 38˚C were only stable until the 3rd day. |
| **MARQUES B, 2012** [54] | Rmt (20˚C to 26˚C), 2˚C to 8˚C, -20˚C | 117 | DBS samples (anti-HCV marker) stored at 2˚C to 8˚C and at -20˚C were stable until the 117th day. However, the samples (anti-HCV marker) that were stored at Rmt were stable until the 60th day, followed by a reduction in antibody concentration on the 117th day. |
| **MCALLISTER G, 2015** [55] | 37˚C, Rmt (22˚C to 28˚C), 4˚C, -20˚C, -70˚C | 200 | There was a loss of antibody concentration in DBS samples (HBsAg and anti-HBc markers) stored at 37˚C, Rmt and 4˚C by the 14th day. The samples that were stored at 37˚C resulted in HBsAg negative for all three patients on the 200th day. However, the samples (HBsAg and anti-HBc markers) that were stored at -20˚C and at -70˚C were stable until the 200th day—as there was minimal variation. DBS samples (anti-HCV marker) were stable at all the temperatures except at 37˚C until the 200th day. Samples which were stored at 37˚C, resulted in a decline of reactivity by 89% on the 200th day. |

*(Continued)*

**Table 4.** (Continued)

| AUTHOR, YEAR | TEMPERATURE | DAYS | OPTIMUM TEMPERATURE TO STORE DBS FOR MAXIMUM ANTIBODY RECOVERY |
|---|---|---|---|
| **SINGH, P** [56] | 37˚C, 2˚C to 8˚C | 210 | The DBS samples (positive for anti-HEV IgM) that were stored at 37˚C were significantly degraded, as 71.43% of the samples became negative on the 55th day, therefore samples were not further analysed. The samples that were stored at 2˚C to 8˚C were stable when tested on the 65th day as all samples were still anti-HEV IgM positive. However, on the 100th and 130th day, 9.52% and 19.04% of the samples became anti-HEV IgM negative. Aside from this, no IgM fluctuation was observed until the 210th day. |

Abbreviations: [a] anti-HBc: Hepatitis B core antibody; [b] anti-HBs: Hepatitis B surface antibody; [c] DBS: Dried blood spot; [d] EBV: Epstein-barr virus; [e] HBsAg: Hepatitis B surface antigen; [f] HCV: Hepatitis C virus; [g] HIV: Human immunodeficiency virus; [h] IgA: Immunoglobulin A; [i] IgG: Immunoglobulin G; [j] IgM: Immunoglobulin M; [k] OD: Optical density; [l] RMT: Room temperature.

There are several limitations to this systematic review. Firstly, the quality of the studies included in this review were generally low (S1 Table), which precluded a meta-analysis of the data. The filter papers, blood volume collected, size of dried blood spots, elution process and the assays used for antibody quantification also differed amongst the studies, compounding the limited translation of results. Furthermore, the variability of the study designs may have also contributed to the heterogeneity which has restricted direct comparisons and prevented any meta-analysis of data, even of the same pathogen. Secondly, differences in specificity, sensitivity and correlation were noted for different pathogens, suggesting that antibodies against some pathogens in DBS may be less stable than others. Thirdly, the studies did not investigate the effect of humidity on DBS, which is often an issue in sub-Saharan Africa and Asia. Finally, we acknowledge that heterogeneity exists when different cut-off levels are applied between the studies.

Further data are needed to demonstrate the stability of DBS for different pathogens, especially bacteria, under different field transport and storage conditions likely to be encountered in low resource settings, including the effect of high ambient temperature or humidity levels.

Consideration of the use of DBS sampling in clinical vaccine or sero-epidemiological studies will depend on both healthcare setting and available infrastructure. The current lack of guidelines for the adaptation of assays from serum to DBS and on the optimal pre-analytical treatment of specimens makes quality control challenging. With optimal storage, DBS can be a useful adjunct to serological analysis due to their relative simplicity to take and requirements for a less rigorous cold chain, saving time and reducing costs.

## Supporting information

**S1 Table. Risk of bias in included studies.**
(DOCX)

**S1 Method. Search strategy.**
(DOCX)

**S2 Method. Prisma checklist for systematic review and meta-analysis.**
(DOCX)

## Author Contributions

**Conceptualization:** Fahimah Amini, Erick Auma, Yingfen Hsia, Paul T. Heath, Kirsty Le Doare.

**Funding acquisition:** Kirsty Le Doare.

**Investigation:** Fahimah Amini.

**Methodology:** Fahimah Amini, Erick Auma, Yingfen Hsia, Sam Bilton, Kirsty Le Doare.

**Supervision:** Kirsty Le Doare.

**Writing – original draft:** Fahimah Amini, Erick Auma.

**Writing – review & editing:** Fahimah Amini, Erick Auma, Yingfen Hsia, Sam Bilton, Tom Hall, Laxmee Ramkhelawon, Paul T. Heath, Kirsty Le Doare.

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
