## [Decision Letter · Decision Letter 0]

24 Sep 2020

PONE-D-20-15259

Reliability of dried blood spot (DBS) cards in antibody measurement: a systematic review

PLOS ONE

Dear Dr.Fatimah Amini

Thank you for submitting your manuscript to PLOS ONE. After careful consideration, we feel that it has merit but does not fully meet PLOS ONE’s publication criteria as it currently stands. Therefore, we invite you to submit a revised version of the manuscript that addresses the points raised during the review process.

We look forward to receiving your revised manuscript.

Kind regards,

William Anderson Paxton, PhD, DIC

Academic Editor

PLOS ONE

Journal Requirements:

2. Thank you for stating in your methods "We searched the electronic databases Embase, Medline and Cochrane library for studies published between January 1st , 1990 and April 30th, 2019". Please also report the last date the databases were accessed.

"I have read the journal's policy and the authors of this manuscript have the following competing interests: KLD has received funding from IMmunising PRegnant women and INfants neTwork (IMPRINT), funded by the GCRF Networks in Vaccines Research and Development which was co-funded by the MRC and BBSRC, the National Vaccine Program Office (NVPO), Bill & Melinda Gates Foundation, Grant OPP1119788, Thrasher Foundation 12250 and NIHR Imperial Biomedical Research Centre KLD2017. KLD has received an honorarium to give a seminar at Pfizer Inc. PTH is an occasional advisor to Pfizer and GSK vaccines. The remaining authors have no competing interests to declare."

4. We note that Figure 2 in your submission contain map images which may be copyrighted. All PLOS content is published under the Creative Commons Attribution License (CC BY 4.0), which means that the manuscript, images, and Supporting Information files will be freely available online, and any third party is permitted to access, download, copy, distribute, and use these materials in any way, even commercially, with proper attribution. For these reasons, we cannot publish previously copyrighted maps or satellite images created using proprietary data, such as Google software (Google Maps, Street View, and Earth). For more information, see our copyright guidelines: http://journals.plos.org/plosone/s/licenses-and-copyright.

4.1.    You may seek permission from the original copyright holder of Figure 2 to publish the content specifically under the CC BY 4.0 license. 

4.2.    If you are unable to obtain permission from the original copyright holder to publish these figures under the CC BY 4.0 license or if the copyright holder’s requirements are incompatible with the CC BY 4.0 license, please either i) remove the figure or ii) supply a replacement figure that complies with the CC BY 4.0 license. Please check copyright information on all replacement figures and update the figure caption with source information. If applicable, please specify in the figure caption text when a figure is similar but not identical to the original image and is therefore for illustrative purposes only.

Reviewers' comments:

Reviewer's Responses to Questions

**Comments to the Author**

1. Is the manuscript technically sound, and do the data support the conclusions?

Reviewer #1: Yes

Reviewer #2: Yes

2. Has the statistical analysis been performed appropriately and rigorously? 

Reviewer #1: N/A

Reviewer #2: N/A

3. Have the authors made all data underlying the findings in their manuscript fully available?

Reviewer #1: Yes

Reviewer #2: Yes

4. Is the manuscript presented in an intelligible fashion and written in standard English?

Reviewer #1: Yes

Reviewer #2: Yes

5. Review Comments to the Author

Reviewer #1: The present study aims to assess the evidence for the use of DBS to accurately measure

antibody concentrations from natural exposure and vaccination. The data is well presented, but some issues should be answered:

- The period of study collection was until April 2019, more than 1 year ago. I recommend to make a new search to include new data.

- Few key words were used to search these papers. It would be interesting to include other synonyms. For example, there are several papers used for DBS collection and not only Guthrie card.

- It was not clear why 16 papers were irrelevant to include in the qualitative analysis.

Reviewer #2: The article by Amini et al, is a systematic review describing the use of dried blood spot cards to measure antibody responses against viral infections. The review is well written and has merit for publication as Dried body fluid spots is an easy and convenient means for diagnostic purposes and it gives access to many areas of the world with limited resources for sample collection and storage.

The reviewer feels that “dried blood spot” on the title should be more precise as dried spot on paper or dried body fluid spot as DBS often refers to whole blood.

The introduction is quite informative and exhaustive however the missing element is the use of such sample vehicle for the follow up of viral load during infection (lines 76...). Plethora of studies have been published such (ayele et al/ DOI: 10.1128/JCM.01919-06, Mwaba et al/ DOI: 10.1016/S0140-6736(03)15103-3 , Brambilla et al/ DOI: 10.1128/jcm.41.5.1888-1893.2003 , Cassol et al/ DOI: 10.1385/0-89603-369-4:139 to name a few.)

In the results section lines 148- 152, the reader would find it easier if this section is expanded with a better description of the figure 1.

Although the review is of great importance for the field and is broad and sound, covering the antibody recovery from dried spot against several infections, it would be stronger if a meta-analysis was performed. The authors provide some explanation as of why that is not done. Can the authors comment whether analysis on individual pathogens is possible?

6. PLOS authors have the option to publish the peer review history of their article (what does this mean?). If published, this will include your full peer review and any attached files.

Reviewer #1: No

Reviewer #2: No

---

## [Author Response · Author response to Decision Letter 0]

9 Feb 2021

Dear Colleagues,

We thank you for your review, we have addressed all of the editor and reviewer comments point by point below and believe the manuscript is much improved as a result.

1. Journal Requirements:

Our manuscript meets the PLOS ONE style requirements

2. Thank you for stating in your methods "We searched the electronic databases Embase, Medline and Cochrane library for studies published between January 1st , 1990 and April 30th, 2019". Please also report the last date the databases were accessed.

This has been added to the methods Page 4 line 102.

This has been added, page 27, line 360

Please confirm that this does not alter your adherence to all PLOS ONE policies on sharing data and materials, by including the following statement: "This does not alter our adherence to PLOS ONE policies on sharing data and materials

This statement has been added, page 20, line 368

4. We note that Figure 2 in your submission contain map images which may be copyrighted. All PLOS content is published under the Creative Commons Attribution License (CC BY 4.0), which means that the manuscript, images, and Supporting Information files will be freely available online, and any third party is permitted to access, download, copy, distribute, and use these materials in any way, even commercially, with proper attribution. For these reasons, we cannot publish previously copyrighted maps or satellite images created using proprietary data, such as Google software (Google Maps, Street View, and Earth). For more information, see our copyright guidelines

We have removed this figure.

Reviewers' comments:

Reviewer #1: The present study aims to assess the evidence for the use of DBS to accurately measure

antibody concentrations from natural exposure and vaccination. The data is well presented, but some issues should be answered:

- The period of study collection was until April 2019, more than 1 year ago. I recommend to make a new search to include new data.

We agree with the reviewer and we have updated the search using the same search terms covering the period from 30th April 2020 until 15th October 2020. We have added an additional 9 articles.

- Few key words were used to search these papers. It would be interesting to include other synonyms. For example, there are several papers used for DBS collection and not only Guthrie card.

The search terms we used to identify potential papers included synonyms for Guthrie card. These included: ‘Guthrie card’, ‘Dried blood spot’, ‘Filter paper’ and mesh term [Antibody]. We believe that these search terms have captured all studies of interest. The full search terms are available in supplementary methods S2.

- It was not clear why 16 papers were irrelevant to include in the qualitative analysis.

We have now added this information to Figure 1.

Reviewer #2: The article by Amini et al, is a systematic review describing the use of dried blood spot cards to measure antibody responses against viral infections. The review is well written and has merit for publication as Dried body fluid spots is an easy and convenient means for diagnostic purposes and it gives access to many areas of the world with limited resources for sample collection and storage.

The reviewer feels that “dried blood spot” on the title should be more precise as dried spot on paper or dried body fluid spot as DBS often refers to whole blood.

The term dried blood spot is commonly used across all journals and studies, especially in regard to establishing disease status in participants. Therefore, we feel that our title accurately reflects the results of original articles included in this systematic review. 

The introduction is quite informative and exhaustive however the missing element is the use of such sample vehicle for the follow up of viral load during infection (lines 76...). Plethora of studies have been published such (ayele et al/ DOI: 10.1128/JCM.01919-06, Mwaba et al/ DOI: 10.1016/S0140-6736(03)15103-3 , Brambilla et al/ DOI: 10.1128/jcm.41.5.1888-1893.2003 , Cassol et al/ DOI: 10.1385/0-89603-369-4:139 to name a few.)

We would like to thank the reviewer for their suggestions. We appreciate that studies have investigated viral load measurement on dried blood spots. However, we feel that as the focus of this review is on immunological readouts (antibodies) rather than viral DNA, means that the inclusion of viral load studies would be confusing for the non-expert reader.

In the results section lines 148- 152, the reader would find it easier if this section is expanded with a better description of the figure 1.

Thank you for your suggestion, we have added further information page 6, line 147.

Although the review is of great importance for the field and is broad and sound, covering the antibody recovery from dried spot against several infections, it would be stronger if a meta-analysis was performed. The authors provide some explanation as of why that is not done. Can the authors comment whether analysis on individual pathogens is possible?

We were very keen to carry out a meta-analysis for the included full text articles, however due to the high heterogeneity of study design, participants and outcomes, we were only able to conduct a narrative synthesis of included studies, summarising the findings with respect to each infectious disease. We have discussed the data with a statistician, who feels that no individual pathogen has data for which a meta-analysis is possible.

Kind regards,

Fahimah Amini (Mres)

---

## [Editor Report · Decision Letter 1]

23 Feb 2021

Reliability of dried blood spot (DBS) cards in antibody measurement: a systematic review

PONE-D-20-15259R1

Dear Dr. Amini,

We’re pleased to inform you that your manuscript has been judged scientifically suitable for publication and will be formally accepted for publication once it meets all outstanding technical requirements.

Kind regards,

William Anderson Paxton, PhD, DIC

Academic Editor

PLOS ONE

Additional Editor Comments (optional):

All points have been adequately addressed
---

## [Editor Report · Acceptance letter]

26 Feb 2021

PONE-D-20-15259R1 

Reliability of dried blood spot (DBS) cards in antibody measurement: a systematic review 

Dear Dr. Amini:

I'm pleased to inform you that your manuscript has been deemed suitable for publication in PLOS ONE. Congratulations! Your manuscript is now with our production department. 

Kind regards, 

on behalf of

Professor William Anderson Paxton 

Academic Editor

PLOS ONE